# A Hypovirulence-Associated Partitivirus and Re-Examination of Horizontal Gene Transfer Between Partitiviruses and Cellular Organisms

**DOI:** 10.3390/ijms26083853

**Published:** 2025-04-18

**Authors:** Ting Ye, Han Li, Du Hai, Zhima Zhaxi, Jie Duan, Yang Lin, Jiatao Xie, Jiasen Cheng, Bo Li, Tao Chen, Xiao Yu, Xueliang Lyu, Xueqiong Xiao, Yanping Fu, Daohong Jiang

**Affiliations:** 1State Key Laboratory of Agricultural Microbiology, Huazhong Agricultural University, Wuhan 430070, China; 18507124815@163.com (T.Y.); lihanhan@webmail.hzau.edu.cn (H.L.); haidududu@163.com (D.H.); drolma@webmail.hzau.edu.cn (Z.Z.); duanjie@webmail.hzau.edu.cn (J.D.); jiataoxie@mail.hzau.edu.cn (J.X.); jiasencheng@mail.hzau.edu.cn (J.C.); boli@mail.hzau.edu.cn (B.L.); taochen@mail.hzau.edu.cn (T.C.); xiaoyu@mail.hzau.edu.cn (X.Y.); xuelianglyu@mail.hzau.edu.cn (X.L.); xueqiongxiao@mail.hzau.edu.cn (X.X.); 2Hubei Key Laboratory of Plant Pathology, College of Plant Science and Technology, Huazhong Agricultural University, Wuhan 430070, China; yanglin@mail.hzau.edu.cn (Y.L.); yanpingfu@mail.hzau.edu.cn (Y.F.); 3National Biopesticide Engineering Research Centre, Hubei Biopesticide Engineering Research Centre, Hubei Academy of Agricultural Sciences, Wuhan 430064, China

**Keywords:** *Sclerotinia sclerotiorum*, mycoviruses, hypovirulence, horizontal gene transfer, partitivirus, coat protein, RdRP, cellular evolution

## Abstract

Previous research has unearthed the integration of the coat protein (*CP*) gene from alphapartitivirus into plant genomes. Nevertheless, the prevalence of this horizontal gene transfer (HGT) between partitiviruses and cellular organisms remains an enigma. In our investigation, we discovered a novel partitivirus, designated Sclerotinia sclerotiorum alphapartitivirus 1 (SsAPV1), from a hypovirulent strain of *Sclerotinia sclerotiorum*. Intriguingly, we traced homologs of the SsAPV1 CP to plant genomes, including *Helianthus annuus*. To delve deeper, we employed the CP and RNA-dependent RNA polymerase (RdRP) sequences of partitiviruses as “bait” to search the NCBI database for similar sequences. Our search unveiled a widespread occurrence of HGT between viruses from all five genera within the family *Partitiviridae* and other cellular organisms. Notably, numerous *CP*-like and *RdRP*-like genes were identified in the genomes of plants, protozoa, animals, fungi, and even, for the first time, in an archaeon. The majority of *CP* and *RdRP* genes were integrated into plant and insect genomes, respectively. Furthermore, we detected DNA fragments originating from the SsAPV1 RNA genome in some subcultures of virus-infected strains. It suggested that SsAPV1 RdRP may possesses reverse transcriptase activity, facilitating the integration of viral genes into cellular organism genomes, and this function requires further confirmation. Our study not only offers a hypovirulence-associated partitivirus with implications for fungal disease control but also sheds light on the extensive integration events between partitiviruses and cellular organisms and enhances our comprehension of the origins, evolution, and ecology of partitiviruses, as well as the genome evolution of cellular organisms.

## 1. Introduction

Mycoviruses, viruses that replicate in fungi, are ubiquitous in nature. Among this diverse group, a few induce hypovirulence in plant-pathogenic fungi, thereby offering a promising avenue for the biological control of plant fungal diseases. A prime illustration is the employment of Cryphonectria hypovirus 1 (CHV1) to mitigate chestnut blight, a devastating disease caused by *Cryphonectria parasitica* [1]. Similarly, Sclerotinia sclerotiorum hypovirulence-associated DNA virus 1 (SsHADV1) transforms its host, the necrotrophic pathogen *Sclerotinia sclerotiorum*, into a beneficial endophyte within plants. This transformation suppresses pathogenic virulence, bolsters plant resistance, fosters growth via the endophytic growth of the SsHADV1-infected strain and makes it a potential plant vaccine [2]. The discovery of hypovirulence-associated viruses has ignited a fervor among phytopathologists to explore mycovirus diversity and potential in biocontrol. Recent advancements in mycovirus research have unveiled new strategies to harness these viruses for combating plant diseases [3,4,5,6].

Viruses in the family *Partitiviridae* have bisegmented or trisegmented double-stranded RNA (dsRNA) genomes, comprising dsRNA1 and dsRNA2 segments, typically ranging from 1300 to 2500 base pairs in length. dsRNA1 encodes an RNA-dependent RNA polymerase (RdRP), while dsRNA2 encodes a coat protein (CP). The partitiviruses virions exhibit isometric, non-enveloped structures, measuring approximately 25–43 nanometers in diameter. Partitiviruses have been reported to infect plants, fungi, and protozoa, and are currently classified into five genera: *Alphapartitivirus*, *Betapartitivirus*, *Gammapartitivirus*, *Deltapartitivirus*, and *Cryspovirus* [7]. However, the taxonomic landscape of partitiviruses is continually evolving, with the recent proposal of two novel genera—Epsilonpartitivirus and Zetapartitivirus—highlighting the existence of numerous unclassified members within this family [8,9,10,11].

While most partitiviruses infect hosts latently, a minority exhibits more dynamic interactions, reducing virulence or altering colony morphology. For instance, Aspergillus fumigatus partitivirus 1 (AfuPV1) induces an abnormal phenotype and attenuates the host growth rate [12]. Similarly, Heterobasidion RNA virus 3-ec1 and Heterobasidion RNA virus 6-ab6 exhibit either beneficial or detrimental effects on *Heterobasidion* spp., depending on the prevailing ecological conditions [13]. Notably, Sclerotinia sclerotiorum partitivirus 1 (SsPV1) imparts hypovirulence to its host [14]. Recently, Colletotrichum alienum partitivirus 1 (CaPV1) has been found to confer hypovirulence to *Colletotrichum alienum* and other species within the genus, expanding the horizons of partitivirus-mediated disease control [15].

Non-retroviral RNA virus sequences (NRVSs) has been consistently documented across the genomes of diverse organisms, spanning fungi, invertebrates, vertebrates, and plants [16,17]. Analogs of the capsid protein (CP) gene from alphapartitiviruses have also been detected in plant genomes, hinting at intriguing genetic interactions. Previous studies have shown that the CP of Sclerotinia sclerotiorum partitivirus S (SsPV-S) reveals 26% amino acid sequence homology with the *Arabidopsis thaliana*’s IAA-leucine-resistant protein 2 (AtILR2) [16]. Similarly, the CP of Rosellinia necatrix partitivirus 2 (RnPV2) exhibits high sequence similarity to AtILR2 [17], and Arabidopsis halleri partitivirus 1 (AhPV1) displays over 50% homology [18]. These findings underscore frequent horizontal gene transfer (HGT) from alphapartitiviruses to plants, yet the specific boundaries and mechanisms of this phenomenon are largely unknown.

*Sclerotinia sclerotiorum* (Lib.) de Bary, a notorious necrotrophic pathogen, is renowned for causing Sclerotinia stem rot on over 700 plant species across 75 families [19,20]. This pathogen is a hotspot for various mycoviruses, including those associated with hypovirulence [21,22]. A recent RNA_seq analysis of *S. sclerotiorum* strains isolated from sunflower fields uncovered a contig indicative of a novel virus from the family *Partitiviridae* [23]. Here, we explore the characteristics of this novel double-stranded RNA virus, designated Sclerotinia sclerotiorum alphapartitivirus 1 (SsAPV1), isolated from the hypovirulent strain AHS232, which harbors multiple viruses. Our analysis encompasses the genome structure, phylogenetic affiliations, and viral influence on its host. Additionally, we identified homologs of *RdRP* and *CP* genes from partitiviruses in the genomes of plants, insects, and prokaryotes, suggesting a potentially widespread occurrence of HGT between partitiviruses and cellular organisms.

## 2. Results

### 2.1. Biological Characteristics and Viruses of Strain AHS232

Under laboratory conditions, strain AHS232 exhibited distinct biological traits compared to strain Ep-1PNA367. After 7 d-cultivation on a PDA plate, it produced profuse hyphae but did not induce lesions on detached rapeseed leaves at 36 hpi (Figure 1A,B). Meanwhile, the growth rate of AHS232 averaged 12.71 mm per 12 h, only half that of Ep-1PNA367 (24.65 mm/12 h) (Figure 1C). RT-PCR verification showed that nine mycovirus species were detected in strain AHS232, including four mitoviruses, Sclerotinia sclerotiorum mitovirus 6 (SsMV6) [24], Sclerotinia sclerotiorum mitovirus 9 (SsMV9) [21], Sclerotinia sclerotiorum mitovirus 14 (SsMV14) [21], and Macrophomina phaseolina mitovirus 1 (MpMV1) [21]; three ourmia-like viruses, Sclerotinia sclerotiorum ourmia-like virus 12 (SsOlV12) [25], Sclerotinia sclerotiorum ourmia-like virus 20 (SsOlV20) (Direct Submission, MW454912.1), and Sclerotinia sclerotiorum ourmia-like virus 7 (SsOlV7) [25]; one endornavirus named as Sclerotinia sclerotiorum endorna-like virus 1 (SsElV1) [23]; and one novel alphapartitivirus related to plant-infecting viruses, tentatively named SsAPV1 (Figure 1D).

### 2.2. Genome Characteristics and Virion of SsAPV1

The genome of SsAPV1 comprises two double-stranded RNA (dsRNA) segments, designated as dsRNA1 and dsRNA2, and an extra truncated segment derived from dsRNA1, termed as dsRNA1-S. dsRNA1 is 1983 bp in length and encodes RNA-dependent RNA polymerase (RdRP) with a predicted reverse transcriptase (RT)-like domain and a “GDD” motif. The RdRP consists of 592 amino acids (aa) with a calculated molecular mass of 69 kDa. dsRNA2, 1805 bp long, encodes a CP of 492 aa, 54 kDa (Figure 2A). The 5′-untranslated region (UTR) of two segments are highly conserved (≥70% identity), while the 3′-UTRs of two dsRNAs are highly similar particularly in adenine-rich domains and both ploy (A) tails were interrupted by a nucleotide (Figure 2B). Full sequences of the dsRNA segments were uploaded to GenBank under the accession number OP080648 and OP080649, respectively.

Two cDNA fragments of dsRNA1 were detected in AHS232 via RT-PCR using a single pair of viral-specific primers. We hypothesized that the smaller segment, dsRNA1-S, was a defective RNA of dsRNA1 (Figure 2C). Sequencing revealed that dsRNA1-S was 1757 bp in length, identical to dsRNA1 except for a 226 bp deletion in the central region of ORF1. This deletion caused a frameshift and the formation of an internal terminator, leading to premature termination during RdRP-ORF translation. Thus, dsRNA1-S does not encode a functional RdRP (Figure 2A).

The virions appeared as spherical particles with an approximate diameter of 26 nm under a microscope. The other eight viruses have no coat protein genes, which are vital to form virion structures, and the morphology of virions met the standards of partitivirus virions. An SDS-PAGE analysis of purified virion proteins showed a 54 kDa band, consistent with the predicted size of SsAPV1 CP (Figure 2D). Thus, they were proposed as the virions of SsAPV1.

To ascertain the relationship between SsAPV1 and other partitiviruses, a phylogenetic tree was constructed using selected viral RdRPs in *Partitiviridae*. The tree shows that the RdRP of SsAPV1 clusters with seven plant viruses within *Alphapartitivirus*. In contrast, the RdRPs of other mycoviruses, such as Botrytis cinerea partitivirus 2 (BcPV2) and SsPV-S (which shares the same host species as SsAPV1), showed a more distant relationship with SsAPV1 RdRP. Therefore, SsAPV1 belongs to *Alphapartitivirus* and is more closely related to plant partitiviruses (Figure 3).

### 2.3. The Transfection of SsAPV1 and Transfectants

Two transfected strains, ATAPV1T2 and ATAPV1T3, were obtained via protoplast transfection (Figure 4A,B and Appendix A). The derived subcultures exhibited similar colony morphology to AHS232, producing abundant mycelium on a PDA plate at 7 dpi (Figure 4A). To confirm biomass difference, the mycelia of these strains were harvested at 6 dpi before sclerotia formation. The mycelial biomass of ATAPV1T2 and ATAPV1T3 was measured at 0.132 and 0.134 mg·mm^−2^, respectively, which is significantly higher than that Ep-1PNA367 (0.084 mg·mm^−2^) and AHS232 (0.107 mg·mm^−2^) (Figure 4C). This suggests that SsAPV1 can improve hosts’ hyphal growth. The growth rates of the two derived strains were both similar to that of Ep-1PNA367 (Figure 4D). The virulence test showed that both ATAPV1T2 and ATAPV1T3 induced small lesions on detached rapeseed leaves. Compared to Ep-1PNA367, the lesions’ sizes caused by ATAPV1T2 and ATAPV1T3 were decreased by 65.9% and 62.3%, respectively (Figure 4E,F), suggesting that SsAPV1 confers hypovirulence to *S. sclerotiorum*.

### 2.4. The Potential RT Activity of SsAPV1 RdRP

By searching the NCBI database, a conserved domain named “RT_like” was annotated in SsAPV1 RdRP, spanning amino acids 196–527 (Figure 2A). To elucidate whether this RdRP possesses reverse transcriptase (RT) activity, DNA templates extracted from infected strains were used to amplify the RdRP and CP genes of SsAPV1. The results revealed that neither CP nor RdRP was amplified from AHS232, whereas amplification was achieved in certain subcultures of the SsAPV1-infected strain, notably subcultures ATAPV1T2 and ATAPV1T3 (marked with a red asterisk in Appendix A). This observation hints at the presence of viral DNA from SsAPV1 in select infected strains and implies that, under certain conditions within fungal cells, the viral RdRP may exhibit reverse transcriptase activity.

### 2.5. Horizontal Gene Transfer of Viral RdRP from Partitiviruses to Other Organisms

Upon searching in NCBI database, we uncovered 52 protein sequences highly similar to partitivirus RdRPs, originating from diverse cellular organisms. These RdRP-like proteins (RdRP-LPs) were disseminated across the genomes of insects, plants, fungi, nematodes, and protozoa, with insects accounting for over half of the identified sequences (Appendix A). The multiple protein alignment of selected partitivirus RdRPs and RdRP-LPs revealed that these proteins share five conserved domains, corresponding precisely to motifs III–VII of the eight conserved domains in partitivirus RdRPs. Notably, these conserved motifs belong to the RT-like domain of viral RdRPs, suggesting potential reverse transcription functions (Appendix A).

To gain further insights, we conducted an evolutionary analysis focusing on RdRP-LPs and RdRPs from selected partitiviruses and partiti-like viruses (Figure 5). The analysis revealed a striking segregation of viral RdRPs and RdRP-LPs from various cellular organisms into 14 distinct clusters (Cluster I–XIV). Notably, Clusters I and II encompassed RdRPs from plant deltapartitiviruses and plant RdRP-LPs. Cluster III was heterogeneous, comprising RdRP-LPs from four insects, one nematodes, and one plants, all closely related to deltapartitivirus RdRPs. Cluster IV was exclusively composed of gammapartitiviruses, while Cluster V harbored mycoviruses in Zetapartitivirus. Cluster VI contained a unique blend: a cryspovirus, three Saccharomyces cerevisiae partitiviruses, and an RdRP-LP from the beetle *Coccinella septempunctata*. In contrast, Cluster VII comprised nine fungal RdRP-LPs related to the RdRP of Cryptosporidium parvum virus 1 (CSpV1).

Intriguingly, our analysis uncovered a plethora of viruses in insects and other arthropods with notable affinities to cryspovirus, such as Atrato Partiti-like virus 6, Enontekio partiti-like virus, etc. [26,27,28,29,30,31]. These partiti-like viruses form two clusters: Cluster VIII comprises two partiti-like viruses and fourteen RdRP-LPs (thirteen from diverse insects and one from a protozoan); and Cluster IX contains nine insect partiti-like viruses and five insect RdRP-LPs. Cluster X embraces viruses proposed as epsilonpartitiviruses, alongside four insect RdRP-LPs. Cluster XI includes two RdRP-LPs, one from *Pomphorhychus laevis* and another from *Eimeria necatrix*. Cluster XII is occupied by betapartitiviruses and a single fungal RdRP-LP. Lastly, Clusters XIII and XIV encompass mycoviruses and plant viruses of *Alphapartitivirus*, alongside two plant RdRP-LPs from *Tanacetum cinerariifolium* and an insect RdRP-LP from *Ceutorhynchus assimilis* (Figure 5). These findings highlight the widespread distribution of RdRP-LPs in plant, insect, nematode, and protozoan genomes, indicative of the frequent horizontal transfer of RdRPs between partitiviruses and cellular organisms.

Furthermore, our investigation revealed that numerous RdRP-LPs were annotated as RNA-dependent RNA polymerases, such as those found in *Amphibalanus Amphitrite* (KAF0302415.1), *Anopheles sinensis* (KFB47171.1), *Choiromyces venosus* (RPA93728.1), *Lolium perenne* (AFA36554.1), *Pyrus pyrifolia* (BAA34783.1), and *Toxocara canis* (KHN74396.1). Moreover, many RdRP-LPs possessed the core domain of RdRP or reverse transcriptase (depicted in Appendix A), implying that these RdRP-LPs may function as RdRPs and RTs within host cells.

### 2.6. Horizontal Gene Transfer of Viral CP from Partitiviruses to Other Cellular Organisms

SsAPV1 CP exhibits significant amino acid (aa) similarity to gene products from various plant and insects and shares over 40% sequence identity with its closest plant protein homologs, consistent with similarities among SsAPV1 CP and other alphapartitivirus CPs. It shows the highest homology (52%) with ILR2 from the common sunflower (the host plant of AHS232) compared to 44% with ILR2 in Arabidopsis thaliana (AtILR2). This exceeds the previously reported 26% similairty between SsPV-S CP and AtILR2 [16]. Intriguingly, the CPs of Rhizoctonia solani partitivirus 1 and Bipolaris maydis partitivirus 1 exhibit an even closer phylogenetic relationship to common sunflower ILR2, with aa similarities of 58% and 54%, respectively (Figure 6).

To determine the evolutionary relationship among these organisms, a phylogenetic tree was constructed, encompassing viral CPs and CP-like proteins (CP-LPs) from diverse cellular hosts. This tree arranges the CPs and CP-LPs in a staggered fashion into twelve distinct clusters. Clusters I through IV are notable for the inclusion of typical alphapartitiviruses alongside 28 CP-LPs from fungi, plants, and insects. Cluster I comprises four plant virus CPs intimately related to ILR2 in *Mercurialis annua* and a hypothetical protein in *Erythranthe guttata*. Cluster II harbors seven plant CP-LPs and viral CPs from SsAPV1, BmPV1, and RosPV1, which display closer kinship to sunflower ILR2 and homologs in *Smallanthus sonchifolius* (*Asteraceae*). Cluster III comprises Arabidopsis halleri partitivirus 1 and seven plant CP-LPs from four eudicot families, particularly *Brassicaceae*. In Cluster IV, plant virus CPs demonstrate a closer affiliation with six plant CP-LPs, while mycovirus CPs are more closely tied to two insect CP-LPs (Figure 7A). Betapartitiviruses and CP-LPs gather in Clusters V and VI. Cluster V harbors a solitary plant CP-LP from zigzag clover, whereas Cluster VI boasts seven insect CP-LPs from two aphids, three beetles, one dragonfly and one plant bug (Figure 7A). Cluster VII includes viral CP from CsPV1 of the *Cryspovirus* genus, Saccharomyces cerevisiae partitiviruses 1–3, and twelve CP-LPs spanning eight fungal origins, three protozoan, and a remarkable find, an ancient Archaea from Asgard group. Intriguingly, CsPV1 CP exhibits the closest association with three protozoan CP-LPs from *Cryptosporidiidae* (Figure 7A).

Cluster VIII solely contains viruses in *Gammapartitivirus*. An outlier, CP-LP (GMN55363.1) from *Ficus carica*, exhibits distant relatedness to gammapartitivirus capsids (Figure 7B). Blastp search revealed two additional CP-LPs, one from a nematode (*Meloidogyne enterolobii*, CAD2197219.1) and another from a mite (*Blomia tropicalis*, KAJ6218664.1) (Appendix A). Moreover, Clusters IX–XII encompass deltapartitiviruses alongside 68 CP-LPs with the majority hailing from eudicot and monocot plants. Eudicots exhibit the greatest diversity, particularly in *Malvaceae* (e.g., *cottons* in Cluster XII), *Asteraceae* (e.g., *sunflower* and lettuce), *Fabaceae* (soybean, green bean, red bean, etc.), and *Brassicaceae* (*Arabidopsis* spp.) in Cluster IX. Monocot CP-LPs stem from *Poaceae*, *Asparagaceae*, and *Orchidaceae*. In addition, several CP-LPs are annotated by the NCBI as functional proteins, including a protein kinase in *Striga asiatica* (GER31701.1), an ABC transporter in *Gossypium australe* (KAA3485722.1), an endonuclease in *Ipomoea trifida* (GLL40948.1), and an F-box-like domain protein in *Arabidopsis suecica* (KAG7557834.1) (Figure 7B).

The phylogenetic tree reveals that plant partitivirus CPs display a closer evolutionary relationship with plant-derived CP-LPs, while mycovirus CPs prefer insect CP-LPs. Another fascinating observation is the coexistence of multiple CP-LPs within a single organism, each sharing sequence similarities with capsids from distinct viral genera. For instance, in *Acer yangbiense*, one CP-LP (TXG73377.1) aligns closely with alphapartitivirus, whereas another (TXG56362.1) is akin to deltapartitivirus (Figure 7A,B). These findings underscore the prevalent integration of the viral gene into plant, insect, and other cellular genomes, occurring independently across various ecological lineages.

To solidify the likelihood of CP-LP integration into the archaeal genome, flanking sequence analysis revealed the presence of transposons and repetitive elements. Specifically, retrotransposons—one non-LTR-containing at the 5′-UTR and another LTR- and non-LTR-containing at the 3′-UTR—were identified. Additionally, two DNA transposons were detected within the CP-LP gene (Figure 8), further supporting the notion of HGT and viral integration in these ancient organisms.

## 3. Discussion

### 3.1. A Novel Hypovirulence-Associated Partitivirus and Its Host Strain

Previous studies have documented diverse mycovirus associated with hypovirulence in *S. sclerotiorum*, and as for partitivirus, there was only SsPV1 in *Betapartitivirus* genus [14]. Herein, we introduce SsAPV1, another alphapartitivirus identified from the multi-virally infected strain AHS232, and confirm its role in inducing hypovirulence in its host. This marks the first report of an alphapartitivirus associated with hypovirulence in *S. sclerotiorum*. While most mycoviruses linked to hypovirulence are known to suppress host growth, SsAPV1 exhibits a unique trait of enhancing the vegetative growth of its host. This characteristic could hold promise for innovative approaches in managing crop stem rot and underscores the need for the further exploration of its potential in biological control.

*S. sclerotiorum* strains are usually co-infected by multiple viruses in nature. Here, nine viruses including SsAPV1 jointly infected the host strain AHS232, which presented abnormal colony morphology and the loss of its pathogenicity. The virion transfection test revealed that virus SsAPV1 has a direct negative impact on host pathogenicity, but the loss pathogenicity was alleviated in transfected strains. Therefore, we assumed that viral co-infection and interaction led to an abnormal phenotype of AHS232. Other eight participants are generally considered to be harmless, so it is worth exploring how SsAPV1 collaborates with these latent ones to threaten the host. Associated studies should contribute to a better application of mycoviruses in the biocontrol of crop Sclerotinia diseases.

### 3.2. Widespread HGT and Transfer Patterns

Present investigations have solely uncovered the integration of capsid protein (CP) genes from alphapartitiviruses within plant genomes [16,17]. However, this study presents a novel paradigm: both *CP* and *RdRP* genes of partitiviruses are integrated into the genomes of various cellular organisms, beyond the confines of alphapartitiviruses and plants, encompassing a broad spectrum of partitiviruses and hosts like plants, fungi, protozoa, animals (predominantly insects), and prokaryotes.

Our findings demonstrate the predominant transfer of *CP* genes from partitiviruses to plants, with insects being the next most frequent recipients, while *RdRP* genes are primarily harbored within insect genomes. RdRP-LPs related to CSpV1 RdRP are exclusively sourced from fungal genomes. Alphapartitiviruses and deltapartitiviruses exhibit a preference for plants as their horizontal gene transfer (HGT) receptors for *CP* genes, while betapartitiviruses favor insects. Three CP-LPs exhibit remote similarity to gammapartitivirus CPs, suggesting the potential integration of gammapartitivirus *CP* genes into cellular organisms. However, the confirmation of this conclusion necessitates a further examination of a wider range of genomes. Notably, a *CP* gene integration event into a protozoan genome was found in the single species of the *Cryspovirus* genus. The CP-LP of Cryspovirus-like viruses (ScPV2 and ScPV3) was identified in an Asgard archaeon genome, the first report of partitivirus gene integration into an archaeal genome. These discoveries underscore the complexity and diversity of viral–host interactions and their implications for genome evolution.

### 3.3. The Origin of Partitivirus

The viral genes that have become integrated into the genomes of cellular organisms are regarded as viral fossils, a testament to the fact that ancestral viruses once infected these cellular hosts. Our findings reveal that ScPV1, ScPV2, and ScPV3 exhibit the closest evolutionary relationship with cryspovirus. Notably, the CPs of ScPV2 and ScPV3 share high sequence similarities (29.9% and 27.9%, respectively) with a gene product (MCP8716971.1) from an Asgard group archaeon. As the Asgard group is believed to be the evolutionary precursor most closely related to eukaryotes [32], the discovery of homologous genes on its genome hints at the antiquity of partitiviruses. Recent investigations have also suggested the presence of partitiviruses in bacteria [33,34], and the remarkable replication of a solitary extant fungal partitivirus across three kingdoms of organisms [35] underscores the possibility that partitivirus ancestors were components of the virome of the last universal common ancestor (LUCA) and may have once been ubiquitous in prokaryotic organisms.

The detection of numerous homologous *RdRP* genes, akin to those found in *Gamma-*, *Delta-*, *Zetapartitivirus*, and *Cryspovirus* within insect genomes spanning diverse families, hints at the once-ubiquitous presence of partitivirus-like entities in insects [26,27,28,29,30,31]. Our further analysis reveals a phylogenetic kinship between these insect RdRP-LPs and partitivirus-like viruses with unclassified fungal partitiviruses, such as Colletotrichum eremochloae partitivirus 1, Rhizoctonia solani dsRNA virus 5, Penicillium aurantiogriseum partiti-like virus 1, and insect partitiviruses (VIII–X), implying that insect-infecting partitiviruses might have originated from their fungal counterparts.

### 3.4. Potential Mechanism and Time Frame for HGT

The reason for the diverse horizontal gene transfer (HGT) frequencies in partitiviruses remains enigmatic. Partitivirus RdRPs and related RdRP-like proteins (RdRP-LPs) harbor a conserved reverse transcriptase (RT) domain. This RT enzyme, crucial in retroviruses and mobile genetic elements [36], synthesizes DNA from RNA. Recent studies indicated that RdRPs in partitiviruses possess RT functionality for DNA synthesis from both homologous and heterologous double-stranded RNA (dsRNA) templates [37]. Our research underscores this point, demonstrating that the potential RT activity of SsAPV1 RdRP may not be detected in all virus-infected strains (Appendix A). Therefore, we propose that the RT activity of partitivirus RdRPs is a pivotal factor contributing to the elevated frequency of HGT from viruses to cellular organisms. This perspective aligns with the assertions made by [37]. While receptors are inclined to integrate exogenous elements, aided by endogenous viral components, it is plausible that the RT function encourages hosts to accommodate partitiviruses with a persistent existence, thereby elucidating why these dsRNA viruses have retained an ostensibly dispensable RT-like domain throughout their lengthy evolutionary journey.

It is astonishing that the precise time frames of these HGT events remain elusive. In CP-LPs, ILR2 and its homologues are prevalent across diverse plant lineages but not necessarily phylogenetically linked to their receptors; remarkably, ILR2 homologues can even be traced to insects (Figure 6 and Appendix A). Hence, we postulate that the progenitors of partitiviruses may have imparted genetic material to their hosts prior to plant group divergence, with independent integration events. The discovery of partitivirus homologous genes in brewing yeast relatives genomes (Figure 7) fortifies our hypothesis. The RT activity of RdRP and partitivirus DNA allude to the potential for HGT to occur swiftly within cells. However, the evolution of these receptor cells into organisms capable of reproducing and subsequently being discovered by humans necessitates the fulfillment of additional criteria; according to genetic bottleneck theory, the horizontal acquired gene should endow special functions for the receptors to survive under lethal adversity, such as resistance against widespread destructive viral infections.

### 3.5. Functional Diversification and Genome Evolution

It is plausible that the integration of partitivirus genes fosters the emergence of novel functional genes within host genomes. Previously, CP-LPs were identified as modulators of indole-3-acetic acid (IAA) conjugate sensitivity and metal transport in *A. thaliana* [38]. Here, we found some CP-LPs are annotated as ATP-binding cassette (ABC) transporters; for instance, a CP-LP (KAA3485722.1) from *Gossypium australe* is classified as an ABC transporter F family member 1, with homologs detected in other *Gossypium* species and *Hibiscus sabdariffa*. Remarkably, viral CP genes have been observed to fuse with host genes. In *S. asiatica*, the CP-LP (GER31701.1) functions as a protein kinase with 881 amino acids, representing a fusion of a protein kinase and a viral CP. Contrastingly, another protein kinase (GER29491.1) in *S. asiatica*, with high similarity to GER31701.1 but devoid of viral CP, comprises 691 amino acids. Similarly, in *Ipomoea trifida*, a viral CP is fused to the C-terminus of a crossover junction endonuclease mus81, creating a fusion protein (GLL40948.1). Additionally, a viral *CP* gene is appended to the 5′-terminus of a gene encoding a protein harboring an F-box-like domain (KAG7557834.1) in *A. suecica*.

If RdRP-LPs are indeed annotated as RNA-directed RNA polymerases, the RdRP-endowed receptors could potentially augment processes related to double-stranded RNA (dsRNA) synthesis, including the RNAi pathway—a crucial antiviral mechanism. Notably, many RdRP-LPs possess highly conserved domains of reverse transcriptase enzymes (Appendix A), suggesting that these receptors may be predisposed to acquiring novel genes from exogenous sources (e.g., RNA viruses and mobile genetic elements) and evolving their own genomes. Hence, the integration of partitivirus genes may positively contribute to the genomic evolution of cellular organisms. However, a meticulous investigation into the precise functions of these CP-LPs and RdRP-LPs is imperative to fully comprehend their roles.

## 4. Materials and Methods

### 4.1. Fungal Isolates and Biological Characteristics

The *S. sclerotiorum* strain AHS232, which displayed abundant aerial hypha and attenuated pathogenicity, was originally activated from a set of sclerotia collected in sunflower (*Helianthus annuus*) fields in Washington State, America, by Prof Weidong Chen at Washington State University. Ep-1PNA367, a virus-free virulent strain, was used as a control and the virus recipient in the viral transfection experiment; it was obtained from the single ascospore isolation progeny of the virus-infected strain Ep-1PN.

All *S. sclerotiorum* strains were cultured on Potato Dextrose Agar (PDA) at 20 °C and stored on PDA slants at 4 °C. The colony morphology was observed 7 days post inoculation (dpi). Mycelial biomass was recorded 6 dpi cultivated on a 90 mm PDA plate. To evaluate the pathogenicity of the strains, detached leaves of rapeseed (*Brassica napus*) were inoculated with mycelial plugs and incubated at 20 °C and 95% relative humidity conditions. Each treatment had more than three replicates and was conducted at least twice.

### 4.2. RNA and DNA Sample Preparation, RT-PCR, and Nested-PCR Amplification

Strains were cultivated on PDA covered with cellophane at 20 °C and fresh mycelia were collected 36 h post inoculation (hpi) for genomic DNA extraction using the PCABIOS method. The total RNA of the tested strains was extracted as described in [39]. cDNA for RT-PCR validation was synthesized using Easy Script One-Step gDNA Removal and cDNA Synthesis SuperMix (TransGen Biotech, Beijing, China). Specific primers were designed to verify the existence of viruses that were detected by NGS analysis in the samples. Nested-PCR amplifications were conducted to verify the potential reverse transcriptase activity of the RdRP of SsAPV1, primers were designed based on SsAPV1 RdRP and CP nucleotide sequences, and DNA samples of tested strains were used as templates. All primers used in this study are shown in Appendix A.

### 4.3. Cloning, Sequencing, and Analysis of Virus Genome

To acquire the viral full-length sequence, the purified total RNA was ligated with an adaptor primer PC3-T7 (Appendix A) via T4 RNA ligase and reversely transcribed to cDNA following a method described previously by [40]. Then, twice terminal amplification was conducted with PC2 (Appendix A) complementary to the adaptor and two viral sequence-specific primers designed based on the available and proximal regions sequences. The products were cloned into the pMD18-T vector (TaKaRa, Dalian, China) for sequencing after purified by a gel extraction kit (Axygen, New York, NY, USA).

Searches for ORFs and conserved domains in each complete viral sequence were performed using the ORF Finder Program (http://www.ncbi.nlm.nih.gov/orffinder, accessed on 6 October 2024) of the National Center for Biotechnology Information (NCBI) and MOTIF research (https://www.genome.jp/tools/motif/, accessed on 6 October 2024). The genomic structure diagrams were produced through IBS (v1.0) [41] following the manufacturer’s instruction. The heatmap of viral CP and ILR2 proteins was computed as described in [10] and imaged by Rstudio (v1.4). Multiple-sequence alignments were generated by clusterX [42] and optimized by Adobe Illustrator (AI, v2019). A maximum-likelihood tree was built to determine the phylogenetic relationship between newly identified mycovirus and selected viruses. BLASTp analysis was performed to explore viral sequences with high similarity to SsAPV1 RdRP or CP and virus-like sequences in other organisms; RdRP-LPs and CP-LPs of other partitiviruses were retrieved in the same way by using the CP and RdRP sequences of exemplar species from each genus in the family Partitiviridae as “seed” sequences; duplicates and contaminating sequences were screened out; then, the conserved domain was aligned with clusterX and trimmed with Bio-edit (v7.0) [43]; the resultant alignments were further loaded in the ProTest server (v3.3) to calculate the best-fit model and the phylogenetic trees were built using PhyML (v3.1); and the original trees were visualized with ChiPlot (http://www.chiplot.online/, accessed on 31 October 2024) and refined using an AI program.

To detect transposons or repetitive sequences in virus-like proteins, an analysis was conducted including Blastp analysis via NCBI and TEs and repetitive sequence searching through Censor in the GIRI (Genetic Information Research Institute) repbase website (http://girinst.org/censor/index.php).

### 4.4. Viral Particle Extraction, Observation, and Transfection

Viral particle was extracted through ultra-centrifugation and purified using the gradient centrifugation method described in [44] with a sucrose density gradient of 30%, 40%, 50%, and 60%. A transmission electron microscope (Model Tecnai G2, 200 kV, FEI Company, Hillsboro, OR, USA) was applied to observe virions negatively stained with 2% (*w*/*v*) phosphotungstic acid solution (pH 7.4).

Protoplast transfection was implemented by inoculating protoplasts of the virus-free strain Ep-1PNA367 with purified SsAPV1 virions in reference to the method in [44] with modifications. Protoplasts were prepared following the protocol described by [45], mixed gently with 50 μL of purified SsAPV1 particle solution, and maintained on ice for 20–30 min. Next, 800 μL of polyethylene glycol (PEG) solution (60% PEG4000, 50 mM Tris-HCl [pH 7.5] and 50 mM CaCl_2_) was added gently, and the mixture was incubated in dark at 20 °C for 20 min. Protoplasts were subjected to recover on solid regeneration medium at 20 °C for 6–8 days. Regenerated colonies were transferred onto PDA plates for the RT-PCR detection of viral genomic sequences. New isolates ATAPV2 and ATAPV3, infected with SsAPV1, were selected for further research.

## Figures and Tables

**Figure 1 ijms-26-03853-f001:**
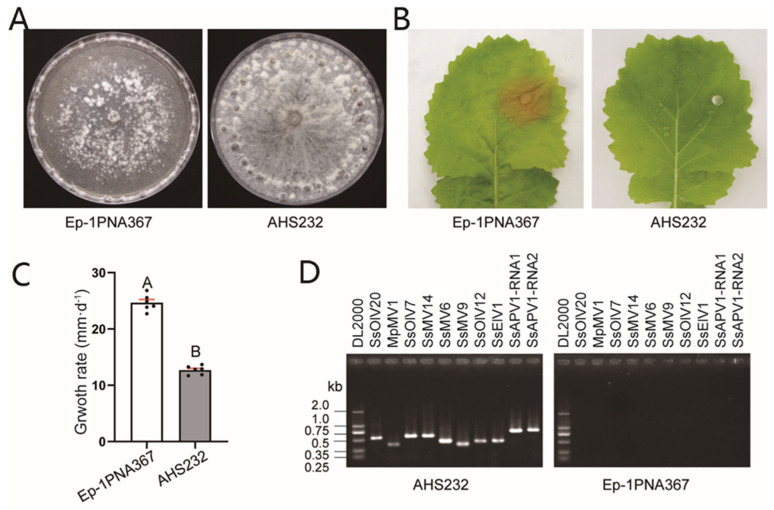
Biological characteristics and viruses in strain AHS232. (**A**) Colony morphology of strains AHS232 and Ep-1PNA367 at 7 dpi. (**B**) Virulence evaluation of each strain on detached leaves of rapeseed (36 hpi). (**C**) Average growth rate assay of strains AHS232 and Ep-1PNA367. Multiple comparison analysis was performed based on Duncan’s multiple range test and different letters on top of each column indicate significant differences (*p* < 0.01). (**D**) RT-PCR amplification to examine existence of RNA mycoviruses in strain AHS232 with specific primer for each mycovirus (Appendix A). Details of viral abbreviations are explained in Appendix A. Lane DL2000 means molecular weight marker DL2000 (TaKaRa, Dalian, China).

**Figure 2 ijms-26-03853-f002:**
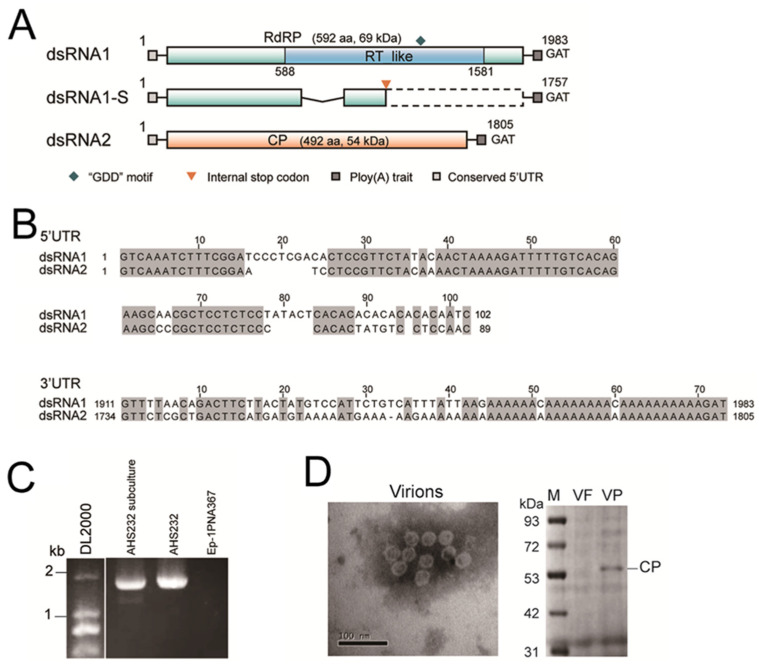
Genome properties and virions characteristics of SsAPV1. (**A**) A diagrammatic sketch of the genome organization of SsAPV1. It contains two dsRNA segments dsRNA1 and dsRNA2 and a defective RNA segment dsRNA1-S. Double-stranded RNA1 encodes a predicted RNA-dependent RNA polymerase (RdRP) with a reverse transcriptase (RT-like) conserved domain. SsRNA2 encodes a putative coat protein (CP). dsRNA1-S possesses a deletion in the central region (the polygonal line). The colored boxes indicate ORFs, while the dashed box indicates a silent region due to an internal terminator (the orange arrowhead). GAT is a terminal base of both dsRNA segments. (**B**) Sequence alignments of 5′-UTRs and 3′-UTRs between two segments of SsAPV1. Identical nucleotides are highlighted in gray. (**C**) An RT-PCR test of dsRNA1 with RdRP-specific primers (Appendix A). Two segments of dsRNA1 are detected in the subcultural generation of strain AHS232. The strain Ep-1PNA367 is used as a control. (**D**) An electron micrograph of purified SsAPV1 virions and CP of SsAPV1 analyzed by SDS-PAGE. VP: SsAPV1 virion solution; VF: the SsAPV1-free strain Ep-1PNA367 fraction at the identical position to SsAPV1 virion solution was used as a control; Lane M means molecular weight standard, PageRulerTM Prestained Protein Ladder.

**Figure 3 ijms-26-03853-f003:**
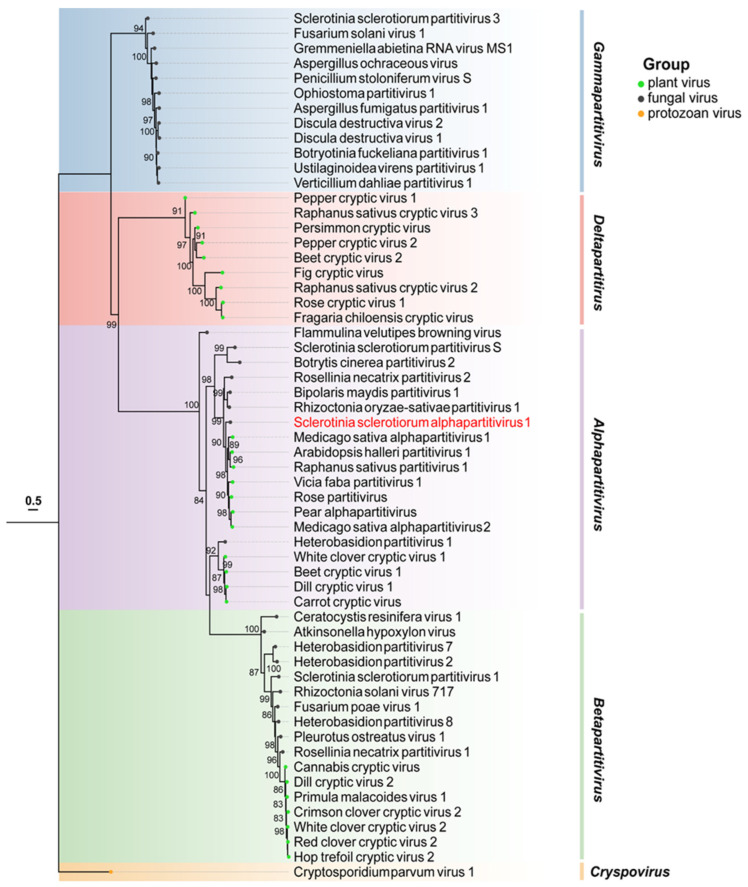
Maximum-likelihood phylogenetic analysis based on RdRP aa sequences of SsAPV1 and selected viruses in *Partitiviridae*. The most appropriate calculating model is VT + I + G + F and branch support values over 80% are noted. Background shading is used to demarcate the five major genera in *Partitiviridae*. Viruses are distinguished by colored dots at the ends of the branches: green is for plant viruses, black is for fungal viruses, and orange is for the virus in protozoan. The aim virus in this paper tagged in red. Abbreviation codes and GenBank accession numbers for these partitiviruses are shown in Appendix A.

**Figure 4 ijms-26-03853-f004:**
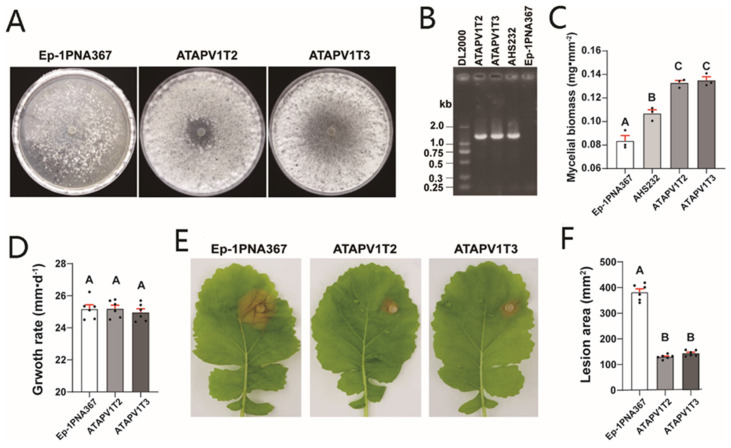
Growth and virulence of SsAPV1-transfected strain Ep-1PNA367. (**A**) Colony morphology of Ep-1PNA367 and two transfectants, ATAPV1T2 and ATAPV1T3 (PDA, 7 dpi). (**B**) RT-PCR test for SsAPV1 with RdRP-specific primers. Strains AHS232 and Ep-1PNA367 are used as positive and negative control, respectively. Lane DL2000 means molecular weight marker DL2000. (**C**) Mycelial biomass of transfectants. Mycelia of these strains were collected from 90 mm PDA plates covered with cellophane before sclerotia formation. Three replicates were conducted for each strain. (**D**) Average growth rate of transfectants. Virus-free strain Ep-1PNA367 was set as control. (**E**) Virulence of transfectants on fresh detached rapeseed leaves. Photos were shot at 36 hpi. (**F**) Lesion area induced by transfectants was calculated by elliptic area formula. Multiple comparison analysis was performed based on Duncan’s multiple range test and different letters on top of each column indicate significant differences (*p* < 0.01).

**Figure 5 ijms-26-03853-f005:**
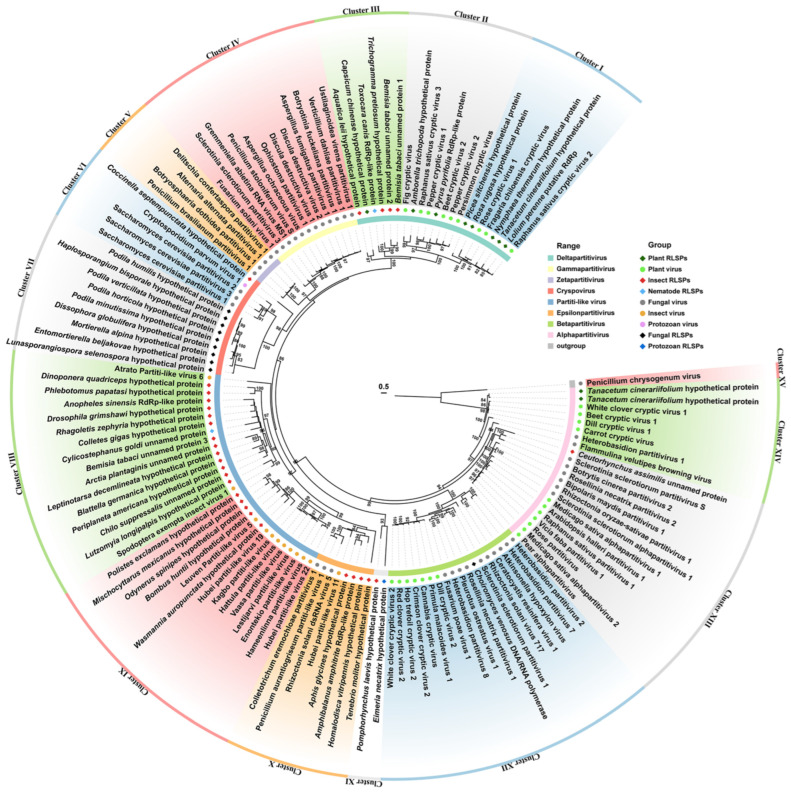
A phylogenetic analysis of partitivirus RdRPs and RdRP-like proteins (RdRP-LPs) in selected cellular organisms with amino acid sequences. A maximum likelihood tree is conducted with the best computed model of VT + R + F, and bootstrap support with values less than 80% were collapsed. Details of RdRP-LPs and GenBank accession numbers are listed in Appendix A.

**Figure 6 ijms-26-03853-f006:**
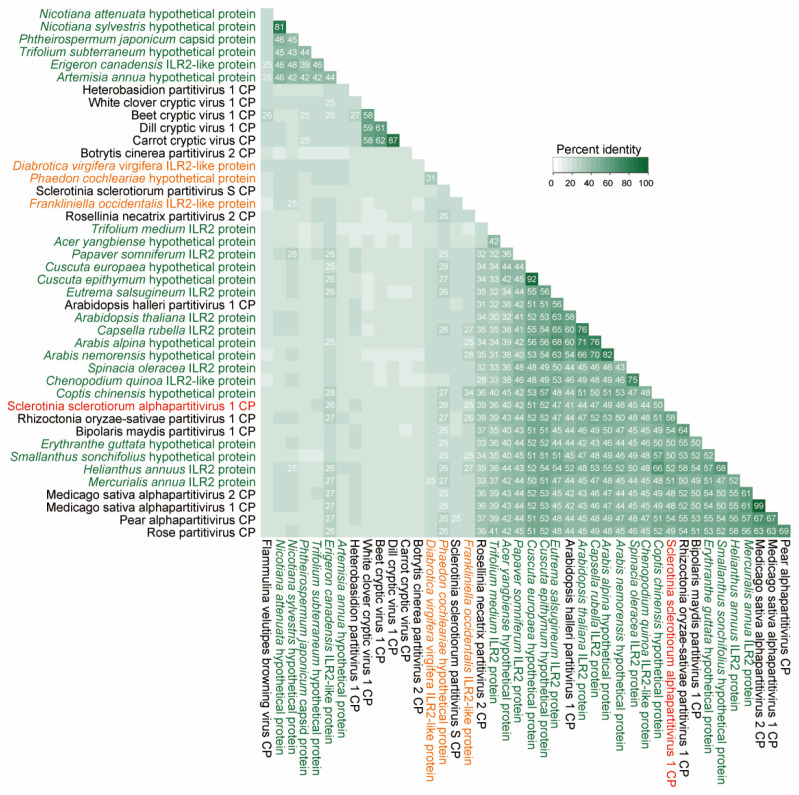
A similarity matrix of the alphapartitiviriuses CPs and CP-like proteins in other organisms. A percent identity matrix was built on amino acid sequences of SsAPV1 (tagged in red), selected CPs of alphapartitiviruses, and proteins in plants (green) or insects (orange).

**Figure 7 ijms-26-03853-f007:**
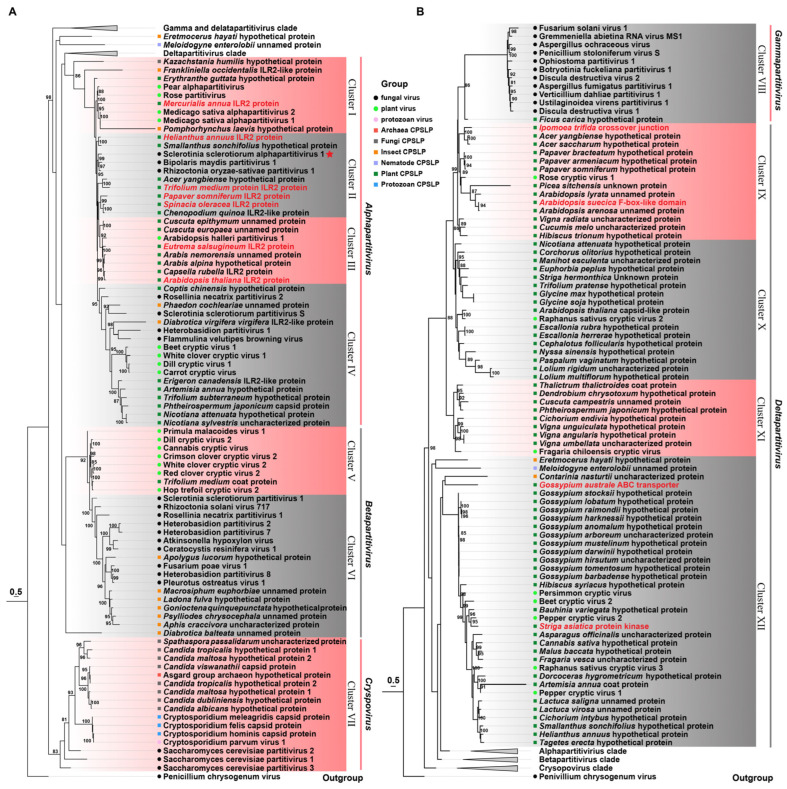
A phylogenetic analysis of viral coat proteins in the family *Partitiviridae* and coat protein-like proteins (CP-LPs) in other organisms. A maximum likelihood tree of selected CPs of partitiviruses and CP-LPs is constructed with the best model of VT + G + F and bootstrap support with values less than 80% collapsed. The tree is presented in two parts of (**A**,**B**). SsAPV1 is marked with a red star, and functional proteins are presented in red. Details of CP-LPs and GenBank accession numbers are listed in Appendix A.

**Figure 8 ijms-26-03853-f008:**
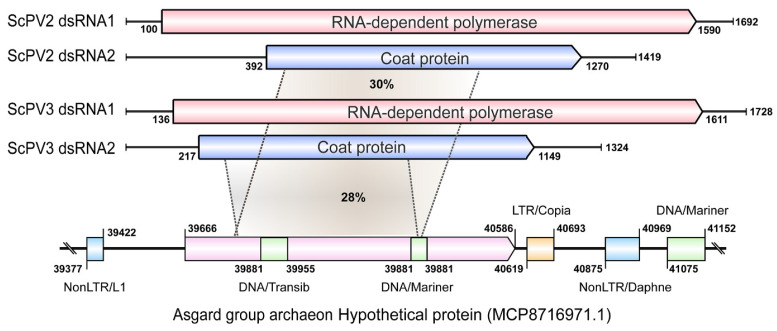
A schematic diagram of the genomes of two CSpV1-like partitivurses and their coat protein-like protein (CP-LP) of the Asgard group archaeon. The homologous regions between CP-LP and CPs of Saccharomyces cerevisiae partitivirus 2 (ScPV2) and Saccharomyces cerevisiae partitivirus (ScPV3) are tagged with gray sectors, and the amino acid identity is presented in each sector. The retrotransposons located on the flanking region of CP-LP are presented with colored rectangular boxes. LTR and nonLTR are LTR-containing retrotransposon and non-LTR-containing retrotransposon. DNA transposons (DNA/Transib and DNA/Mariner) located in the CP-LP and on the far flanking region are also displayed. This analysis is conducted by using Blastp analysis via the NCBI and TEs and repetitive sequence searching through Censor in GIRI (Genetic Information Research Institute) repbase website (http://girinst.org/censor/index.php, accessed on 6 October 2024).

## Data Availability

Data is contained within the article and Appendix A.

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
