# Peer review of "A Hypovirulence-Associated Partitivirus and Re-Examination of Horizontal Gene Transfer Between Partitiviruses and Cellular Organisms"

_ijms, 2025, doi:10.3390/ijms26083853_

Round 1
Reviewer 1 Report
Comments and Suggestions for Authors
The manuscript by Ye et al describes the characterization of a novel Partitivirus 1, Sclerotinia sclerotiorum alphapartitivirus 1 (SsAPV1) and the investigation of horizontal gene transfer (HGT) of Partitivirus genes into cellular organisms by search of genome database with viral coat protein (CP) and RNA-dependent RNA polymerase (RdRP) sequences of Partitiviruses as “baits”. SsAPV1 as a new member of Partitivirus, was analyzed to determine its genome structure, phylogenetic relation to other Partitiviruses, and impact on its host fungus. Of partocular interest is that this virus was able to attenuate the virulence of its host Sclerotinia sclerotiorum strain. They also found that analogs of the Partitivirus coat protein gene (CP) and RNA-dependent RNA polymerase (RdRp) were presence in the genome of many cellular organisms, including plants, protozoa, animals, fungi, and even archaeon. They further showed that DNA fragments originating from the SsAPV1 RNA genome could be amplified by PCR from the fungal genome of strains that were infected by SsAPV1. Thus, the authors proposed that the RdRP of SsAPV1 possesses reverse transcriptase activity to facilitate the integration of viral genes into cellular organism genomes. The findings are of interest and significance, offering a new perspective on genome evolution of the host of Partitivirus, as well as the ecology of virus-host interaction. While the manuscript is in good quality, I have a few concerns.
Majors
- The manuscript is composed of two related stories: characterization of of sclerotiorum alphapartitivirus 1 and investigation of horizontal gene transfer (HGT) of Partitivirus genes into cellular organisms. In my opinion, it is better to split the manuscript into two separate manuscripts.
- The most compelling evidence to support the hypothesis that the reverse transcriptase activity of RdRP of a Partitivirus facilitates horizontal gene transfer between Partitiviruses and cellular organisms, is to demonstrate biochemically that the RdRP of SsAPV1 really possesses reverse transcriptase activity.
Minors
- In the Introduction, the authors emphasize “frequent horizontal gene transfer (HGT) between alphapartitiviruses and plants”. It is better to offer a few examples of horizontal gene transfer from plants into the genome of Partitiviruses, to balance the claim of horizontal gene transfer (HGT) between alphapartitiviruses and plants.
- “2.1. Biological characteristics and viruses of strain AHS232”
Bulleted lists look like this: Under laboratory conditions – Is this paragraph written by AI?
- Please explain why Ep-1PNA367 was selected as a control. Ep-1PNA367 and AHS232 are not isogenic genetically.
- Please define the meanings of RdRP-LS and CP-LS.
- Please rephrase the sentence “To characterize whether virus-like proteins and their flanking sequences in cellular genomes contain transposons or repetitive sequences, analysis has been conduct including Blastp analysis via NCBI and TEs or…”
- Figure 2. Could particles of all nine viruses in strain AHS232 be extracted and separated each other? How could the virions shown in Figure 2D be identified as SsAPV1? Should explain in the text.
- “2.5. Horizontal gene transfer of viral RdRP from partitiviruses to other organisms”
“Upon searching in NCBI database, we uncovered 52 protein sequences highly similar to partitivirus RdRPs, originating from diverse cellular organisms” --- considering the very large number of genome sequences in the NCBI database, 52 protein sequences highly similar to partitivirus RdRPs is in fact, a very small proportion, i.e., a rare event. If the RdRP of SsAPV1 possesses RT activity, AHS232 should be the one most likely to harbor DNA copy of the RdRP or CP in its genome. However, amplicons of RdRP and CP were amplified from ATAPV1T2 and ATAPV1T3, the only two tranfectants, but not from the AHS232. This efficiency seems too high. I would suggest that these DNA template- amplified fragments be sequenced to verified their identity.
- Discussion: compared to SsAPV1, discussion on HGT was much over-weighted.
A few sentences need to be rephrased to better clarify the meaning.
Reviewer 2 Report
Comments and Suggestions for Authors
This MS reports complete nucleotide sequences of both dsRNA genome segments of a novel partitivirus, Sclerotinia sclerotiorum alphapartitivirus 1 (SsAPV1). This virus was isolated from a hypovirulent S. sclerotiorum strain AHS232. This study also reported successful infection of the virus-free S. sclerotiorum strain Ep-1PNA367 with SsAPV1 virus particle preparation isolated from the strain AHS232. Some biological properties of the SsAPV1-transfected lines ATAPV1T2 and ATAPV1T3 were more similar to those of AHS232, rather than their parental Ep-1PNA367. This included S. sclerotiorum colony morphology (Fig. 4A), hyphal growth, and reduced virulence compared to Ep-1PNA367 (Fig 4.E).
In the previous studies (Peyambari et al 2021 Mol. Biol. Evol. doi:10.1093/molbev/msab235 ), showed that RNA-dependent RNA polymerase of partitiviruses possess RNA-dependent-DNA polymerase (reverse transcriptase) activity. It is widely accepted that the reverse transcriptase activity of partitiviruses results in a high incidence of insertions of partitivirus cDNAs in the nuclear genomes of theirs hosts. In this study, a systematic bioinformatic search analysis of partitivirus-derived cDNA inserts (RdRp and CP genes) was also carried out (sections 2.5, and 2.6), which showed a widespread insertions of alphapartitivirus genes in plant and insect genomes.
The results, including experimental characterization SsAPV1 genome and the impact SsAPV1 on S. clerotiorum biology, and bioinformatic analysis of horizontal gene transfer of alphapartitivirus genes, are novel and will be of interest to virologists, mycologists and plant pathologist, but there are several point which should be addressed:
Points to be addressed:
Section 2.3. "The transfection of SsAPV1 and transfectants."
In this study, virus preparation from S. sclerotiorum strain AHS232 was used to transfect virus-free strain Ep-1PNA367 to produce SsAPV1-infected S. sclerotiorum lines ATAPV1T2 and ATAPV1T3. Although 9 mycoviruses, which were detected in AHS232 in addition to SsAPV1, do not form virus particles and unlikely to be present in the virus preparation introduced to Ep-1PNA367 during transfection, I suggest to test the presence of other viruses (as Fig. 1D) in the lines ATAPV1T2 and ATAPV1T3. Results of this analysis should be presented either as a Supplement or as a separate panel in Fig. 4. This would strengthen the suggestion that SsAPV1 is a causal agent of biological properties changes of Ep-1PNA367 after introduction of the SsAPV1 virus.
Section 2.4. "The potential RT activity of SsAPV1 RdRp."
Supplementary Figure S1: Indicate expected positions of the specific PCR fragments (RdRP, CP) with an arrows. The figure for RdRp in Figure S1 does not look convincing.
In this study no analysis was carried out to demonstrate integration of the ScAPC1 into Sclerotinia genome. Potentially, the SsAPV1 -derived cDNA could be present as as episome.
Fig. 4D. (Growth rates)
To include results for AHS232 to this graph.
Fig. 4F. (Lesion areas)
To include results for AHS232 to this graph.
Fig. 5 and Fig. 6.
Specify if nucleotide of amino acid sequences were used.
Legend to Fig. 6.
The phrase "(tagged in red") makes no sense.
Abstract: Line "This revelation suggests that the RdRP of SsAPV1 possesses reverse transcriptase activity, potentially facilitating the integration of viral genes into cellular organism genomes"
This study did not investigated RT activity of SSsAPV1 directly, this suggestion is based on the previous studies (e.g. Peyambari, et al. RdRp or RT, That is the Question. Mol. Biol. Evol. 2021, 38, 5082-5091, doi:10.1093/molbev/msab235.).
Round 2
Reviewer 1 Report
Comments and Suggestions for Authors
The revision has adequately addressed all of my concerns and I am happy with this version.
There is an issue that needs to be clarified:
3.2. Widespread HGT and Transfer Patterns
Present investigations solely uncovered the integration of capsid protein (CP) genes from alphapartitiviruses within plant genomes [16,17].
---- Present investigations or Previous investigations?